# Prospection of Peptide Inhibitors of Thrombin from Diverse Origins Using a Machine Learning Pipeline

**DOI:** 10.3390/bioengineering10111300

**Published:** 2023-11-09

**Authors:** Nivedha Balakrishnan, Rahul Katkar, Peter V. Pham, Taylor Downey, Prarthna Kashyap, David C. Anastasiu, Anand K. Ramasubramanian

**Affiliations:** 1Department of Chemical and Materials Engineering, San José State University, San Jose, CA 95192, USAprarthna.kashyap@sjsu.edu (P.K.); 2Department of Computer Science and Engineering, Santa Clara University, Santa Clara, CA 95053, USAdanastasiu@scu.edu (D.C.A.)

**Keywords:** antithrombotic, anticoagulant, peptide design, classification, regression

## Abstract

Thrombin is a key enzyme involved in the development and progression of many cardiovascular diseases. Direct thrombin inhibitors (DTIs), with their minimum off-target effects and immediacy of action, have greatly improved the treatment of these diseases. However, the risk of bleeding, pharmacokinetic issues, and thrombotic complications remain major concerns. In an effort to increase the effectiveness of the DTI discovery pipeline, we developed a two-stage machine learning pipeline to identify and rank peptide sequences based on their effective thrombin inhibitory potential. The positive dataset for our model consisted of thrombin inhibitor peptides and their binding affinities (*K_I_*) curated from published literature, and the negative dataset consisted of peptides with no known thrombin inhibitory or related activity. The first stage of the model identified thrombin inhibitory sequences with Matthew’s Correlation Coefficient (MCC) of 83.6%. The second stage of the model, which covers an eight-order of magnitude range in *K_I_* values, predicted the binding affinity of new sequences with a log room mean square error (RMSE) of 1.114. These models also revealed physicochemical and structural characteristics that are hidden but unique to thrombin inhibitor peptides. Using the model, we classified more than 10 million peptides from diverse sources and identified unique short peptide sequences (<15 aa) of interest, based on their predicted *K_I_*. Based on the binding energies of the interaction of the peptide with thrombin, we identified a promising set of putative DTI candidates. The prediction pipeline is available on a web server.

## 1. Introduction

Ever since its discovery in 1872, thrombin has occupied the center stage in the pathophysiology of cardiovascular diseases [1]. Over the years, important roles of thrombin have also been identified in the pathophysiology of a multitude of other diseases including cancer, autoimmune and inflammatory disorders, and most recently in COVID-19 [2,3,4]. The most prominent function of thrombin is the conversion of plasma fibrinogen to a crosslinked polymeric fibrin network, the structural and functional unit of the blood clot. The inability to generate sufficient thrombin can result in hemorrhage, while unregulated and excessive thrombin generation can lead to thrombosis and tissue damage. Thrombin is uniquely capable of regulating its own production through positive and negative feedback loops involving other enzymes of coagulation and complement activation cascades [5]. Therefore, molecules that precisely tune thrombin activity have always been both fundamental and clinically relevant and are of interest to academia and pharma [6].

Thrombin activity can be modulated either indirectly by altering thrombin generation rates or directly by interfering with thrombin action. Direct thrombin inhibitors (DTIs) are a new class of anticoagulants that bind directly to thrombin and block its interaction with its substrates [7]. The peptide hirudin and its derivatives, including bivalirudin, lepirudin, and desirudin, are the largest class of DTI and have been approved by the US FDA for the treatment of heparin-induced thrombocytopenia, percutaneous coronary intervention, and for prophylaxis against venous thromboembolism. Small peptidomimetic DTIs (dabigatran and argatroban) have also been in clinical use. DTIs are potent antithrombotic agents as they have a higher capacity for the inhibition of fibrin-bound thrombin than indirect inhibitors, such as thrombin, since bound thrombin can actively promote thrombus growth. They also have shorter half-lives in plasma and do not require cofactors for their activity. Despite these advantages, current DTIs are contraindicated in certain situations or limited in their applicability as they are associated with bleeding or thrombotic risks [8]. Therefore, efforts to discover new antithrombotic agents are necessary to meet the capacious demands of adverse cardiovascular events [9].

To this end, we sought machine learning approaches for the discovery of new peptide inhibitors of thrombin. We are motivated to work with peptide inhibitors of thrombin because: (1) the clinical relevance of hirudin, the most well-known and widely utilized DTI, which is a peptide; (2) the analogs and derivatives of hirudin, particularly bivalirudin, with properties that are significantly more desirable than the parent hirudin; (3) the fundamental importance of thrombin-inhibiting peptides in the survival of hematophagous animals, such as leech, snakes, and ticks. In fact, hirudin was discovered from the saliva of the leech *Hirudo medicinalis*; and (4) the wide range of binding affinities spanning nearly 8-orders of magnitude in existing thrombin-inhibiting peptides suggests that peptide inhibitors offer an opportunity for fine-tuning the inhibitory potential.

When carefully built, ML-based models can rapidly screen a vast chemical and biological space, enabling accurate and faster in silico predictions of the biological activity of new molecules. As a cheminformatics model, ML combines chemistry, computer science, and information technology to aid in drug discovery through tasks like virtual screening, library design, and high-throughput screening analysis [10,11,12]. Machine learning algorithms leverage large chemical datasets for predictive modeling and pattern recognition, including the prediction of the properties and activities of peptides based on their sidechains [13,14,15,16]. This integration has accelerated the discovery and design of novel peptides with desired biological activities, opening new avenues for peptide-based drug development. Recently, machine learning models have been developed for the discovery of novel antimicrobial peptides [17], anticancer peptides [18], antibiofilm peptides [19], antihypertensive peptides [20], and membrane-active peptides [21]. Some of the model predictions have already started to show promise in in vitro and in vivo tests [22,23]. Despite the clinical and fundamental importance of antithrombotic peptides, there have not been any attempts to exploit machine learning for the expedient discovery of new thrombin inhibitors with desired activity levels.

In this work, we develop a computational ML pipeline to predict the thrombin inhibitory activity of the peptides from their properties, which are quantitative structure-activity relationship (QSAR) descriptors. This two-stage pipeline consists of classification models to identify, from a very large peptide database, a handful of hits with thrombin inhibitory activity, and regression models that predict the level of activity of the peptide hits. The models processed the multi-dimensional QSAR properties to prioritize those that are key to thrombin inhibition. The identified hits were then ranked based on their binding affinity to thrombin, as determined by the molecular modeling of their interactions. The prediction pipeline is available on the web server: https://thrombin-inhibitor-peptide-predictor.info (accessed on 3 November 2023).

## 2. Methods

### 2.1. Dataset Preparation

For our study, we collected datasets from various sources including literature, patents, and protein databases.

Positive Dataset. We collected only direct thrombin-inhibiting peptides reported as “antithrombotic” from peer-reviewed publications. We collected the sequences of these peptides from UniProt [24], NCBI Protein Database [25], RCSB PDB [26], and PubChem [27]. Next, we obtained the experimentally determined inhibition constants of these peptides against thrombin, also from peer-reviewed publications. After removing the duplicates, we obtained 88 naturally occurring antithrombotic peptides, and inhibition constants of 53 of these peptides (Appendix A). Only peptides containing naturally occurring amino acids were chosen.Negative Dataset. To prepare the non-antithrombotic negative dataset, we collected peptides from the UniProt and NCBI databases that were not annotated as “anticoagulant”, “antithrombotic”, “hemostasis-impairing”, “antimicrobial”, “anti-inflammatory”, or “thrombin inhibitor”. To minimize bias in the random selection of peptides agnostic to thrombin binding, the ratio of collected negative to positive peptides was 9:1, and we maintained a similar ratio for different sequence lengths within the dataset. We compared the sequences within the negative dataset between the negative and positive datasets for an 80% sequence match and removed the ones above this threshold. Finally, we obtained a negative dataset with 792 sequences.Test dataset. To identify thrombin-inhibiting activity in new peptides, we collected a total of 10,743,304 peptides from the UniProt and NCBI protein databases. We searched for the source organism as one of ‘fungi’, ‘bacteria’, ‘snakes’, ‘leeches’, ‘humans’, ‘mice’, ‘eukaryota’, and ‘viruses’, and the results were filtered to a sequence length between 5 and 200 amino acids. Peptides in the sequence range of 5 to 15 amino acids were collected independently of the source. The peptides could be true peptides or random fragments of large proteins.

### 2.2. Feature Extraction

We extracted features to describe the peptide sequences using global protein sequence descriptors [28]. These features are:Global Physico-Chemical Properties (PCP). We used the ‘Biopython ProtParam’ package to extract the global properties of the collected sequences which include sequence length, molecular weight, aromaticity, isoelectric point, and instability. This constitutes a 5-element vector.Amino Acid Composition (AAC). The amino acid composition (AAC) is a measure that quantifies the relative abundance of each amino acid in a peptide sequence. These features were extracted using the ‘propy3′ python package [29]. The following equation represents the amino acid composition function:
(1)$$\mathrm{AAC}\;(i)=\frac{Total\;number\;of\;amino\;acid\;of\;type\;\left(i\right)}{Total\;number\;of\;amino\;acids}\times 100.$$Composition Transition Distribution (CTD). The CTD descriptor is a 147-element vector that describes different physico-chemical properties of a peptide [30] (Appendix A). The physico-chemical properties covered by CTD features are ‘polarity’, ‘polarizability’, ‘charge’, ‘secondary structure’, ‘hydrophobicity’, ‘normalized van der Waals volume’, and ‘solvent accessibility’. The CTD descriptor groups the amino acids into three classes for each physico-chemical property. The composition (C) descriptor describes the global percentage of each class in a peptide sequence, the transition (T) descriptor characterizes the percent frequency of transitions between two classes in a peptide sequence, and the distribution (D) descriptor specifies the distribution patterns of each class in a sequence. These CTD properties were extracted using the ‘propy3 CTD’ package.Dipeptide Composition (DPC). The DPC descriptor was extracted using the ‘propy3 AAComposition’ package which returns a 400-element vector containing percent fractions of dipeptides, i.e., AA, AC, AD, …, VY, and VV, in a peptide sequence. The DPC fraction percentage is calculated as follows:
(2)$$\mathrm{DPC}\;(i,\;j)=\frac{Total\;number\;of\;dipeptides\;of\;amino\;acid\;of\;type\;\left(i\;and\;j\right)}{Total\;number\;of\;possible\;dipeptides\;available}\times 100.$$

Together, 572 peptide features were extracted which were used for training machine learning models using the 880 peptides in the positive and negative datasets.

### 2.3. Classification Models

We implemented machine learning models for predicting thrombin inhibitory activity, employing various classification algorithms. To evaluate model performance, we randomly split the data into three sets: 60% for training, 20% for validation, and 20% for testing, which ensures an adequate presence of positive samples in all datasets. The training set consisted of 528 samples, the validation set of 176 samples, and the testing set of 176 samples, which served as out-of-sample test data. We employed a support vector machine (SVM) with both linear and radial basis function kernels, logistic regression, random forest, k-nearest neighbors (kNN), and extreme gradient boosting (XGBoost) models for classification. The performance of these models was compared to determine the most suitable one for our final inference. We implemented these models using the widely used scikit-learn package [31], a popular machine learning library in Python.

First, all the baseline models were tuned by choosing the hyperparameters that lead to the best Matthew’s Correlation Coefficient (MCC) score on the validation set. We used the RandomizedSearchCV package for hyperparameter tuning and performed 5-fold cross-validation across the joint training and validation data sets. The SVM models were tuned for hyperparameters *C* and 𝛾; the random forest and XGBoost models for the number of estimators, maximum depth, minimum sample leaf node, and minimum sample split; the logistic regression model was tuned for *C*; and the kNN model for the number of nearest neighbors. The imbalance in the dataset was accounted for by setting the ‘class_weight’ parameter of the classifiers to ‘balanced’ which adjusts the weights according to class frequencies in the dataset. The final models were tested on labeled out-of-sample test sets and their MCC performance was compared with the average MCC score obtained during cross-validation to ensure that the models did not have high generalization errors. Based on these criteria, the best-performing model was chosen and was used to predict thrombin inhibitory activity in peptides collected from protein databases. The performance of the classification models was also estimated using Accuracy and *F*_1_ score (harmonic mean of precision and recall). The three effectiveness measures are defined as:(3)$$\mathrm{Accuracy}=\frac{TP+TN}{TP+FN+TN+FP},$$
(4)$$F_{1}=\frac{TP}{TP+\frac{FP+FN}{2}},$$
(5)$$\mathrm{MCC}=\frac{\left(TP\right)\left(TN\right)-\left(FP\right)\left(FN\right)}{\sqrt{\left(TP+FP\right)\left(TP+FN\right)\left(TN+FP\right)\left(TN+FN\right)}},$$
where *TP* is true positive, *TN* is true negative, *FP* is false positive, and *FN* is false negative.

### 2.4. Clustering

We utilized clustering techniques to group peptides based on similar characteristics. This approach was applied to both the positive peptides in our dataset and the new test peptides collected from protein databases. To ensure uniqueness among the new test peptides, we selected medoids as representatives for each cluster. We employed a hierarchical clustering model to group together similar peptides as determined by the Euclidean distance metric between the feature sets [32]. The cluster proximity metric used was Ward’s linkage, which aims to minimize the variance within each cluster when merging clusters. We used the agglomerative clustering algorithm from the scipy.cluster.hierarchy module [33]. For each threshold value given by the clustering algorithm, the number of clusters and silhouette scores were obtained. The optimal number of clusters was determined based on the highest silhouette score, defined as (*b* − *a*)/max(*a*,*b*), where ‘*a*’ and ‘*b*’ represent the average distance between a data point and all other points within the same cluster, and the average distance between the data point and all points in the nearest neighboring cluster, respectively. We computed the Euclidean distance matrix between peptides within each cluster using the scipy.spatial.distance module. Then, we identified medoids, the peptides with the smallest total distance to all other peptides within their cluster, which capture the characteristics of all the peptides in their respective clusters.

### 2.5. Regression Models

In addition to predicting thrombin inhibitory activity, we also utilized regression models to estimate the inhibition constant (*K_I_*) for the positive peptides. Specifically, we employed three regression models, Support Vector Regressor with radial basis function, Support Vector Regressor with linear kernel, and Lasso regression, which were trained using the sklearn.svm and sklearn.linear_model modules. Out of the 88 positive samples in our dataset, 53 had a *K_I_* value. To evaluate the performance of the regression models, we split the dataset into training, validation, and testing sets following a 60%, 20%, and 20% ratio. As the inhibition constant (*K_I_*) values span a wide range of eight orders of magnitude, we converted them to a logarithmic scale before training the regression models. The best model was selected based on the root mean squared error (RMSE) score, defined as RMSE = sqrt (1/*n* * ∑ (*y* − *y_pred*)^2^), where *n* is the number of data points, *y* is the actual value of the target variable, and *y_pred* is the predicted value of the target variable. To improve the model performance, the hyperparameters of these models were tuned through 5-fold cross-validation using GridSearchCV; the support vector models were optimized for *C*, ε, and 𝛾, while the lasso regression was optimized for 𝛼. To prevent overfitting, we adopted a Sequential Forward Selection (SFS) for feature selection with 5-fold cross-validation to select the optimal set of features. This new set of features was used in all the models. Finally, we selected the regression model with the lowest average validation RMSE score over the 5 folds in the 5-fold cross-validation as the best-performing model, which was used to predict the *K_I_* values of new peptides.

### 2.6. Molecular Docking

For docking peptide sequences with thrombin, we used two web servers, HPEPDOCK [34] and CABS-dock [35]. The PDB file of thrombin (1PPB) and the peptide sequence was entered as inputs to the server utilizing default parameters, and the top-ranked pose of the protein–peptide complex was used for further analysis. The binding strength between the peptide and thrombin was inferred from the docking score estimated by HPEPDOCK. The binding energy was estimated by first obtaining the protein–peptide complex in the PDB format from CABS-dock. Then, the complex structure was entered as input to another server, PRODIGY [36]. Lastly, the determination of the binding sites of the peptide on thrombin was accomplished using PyMol, using a cutoff of 5 Å distance between the interacting atoms.

## 3. Results

The machine learning pipeline to identify potent inhibitors of thrombin consisted of three phases (Figure 1). First, a classification model was built to predict peptides with thrombin inhibitory activity. Second, a regression model was constructed to estimate the *K_I_* values of peptides with thrombin inhibitory activity. Third, a large set of peptides from databases was tested to identify new peptide candidates that exhibit thrombin-inhibiting activity. The predictions of the machine learning model were confirmed by protein docking studies.

### 3.1. Characteristics of Thrombin-Inhibiting Peptides

Thrombin inhibitory peptides are direct thrombin inhibitors (DTI) that block the proteolytic action of thrombin on its substrates by interacting with one or more of the three binding sites, namely, the active site (AS), the Anion Binding Exosite I (ABEI), and the Anion Binding Exosite II (ABEII) [37,38] (Appendix A). The AS cleft is hydrophobic and flanked by the exosites on either side. Recent studies have revealed that thrombin activity is also influenced by the binding of the substrates to two additional exosites, namely, the hydrophilic γ-loop and Na^+^ binding site. Bivalent DTIs, such as hirudin and bivalirudin, bind to both the active site and exosite I of thrombin, which enhances their inhibitory effect on thrombin activity.

From the published literature, we curated 88 DTI peptides and inhibition constants (*K_I_*) for 53 of these peptides. These sequences, the *K_I_* values, and the reference sources are listed in Appendix A, and their 578 features are listed in a Appendix A. The vast majority of the peptides in the positive dataset contained less than 160 amino acids (97.1%), and most of the peptides (82.7%) contained between 3 and 70 amino acids (Figure 2A). As shown in Figure 2B,C, these peptides contain higher percentages of E (13.6%), G (9.1%), D (8.7%), and P (8.7%), and lower percentages of W (0.34%), M (0.7%), H (1.9%), Y (3.6%), Q (3.7%) and I (3.9%). C is over-represented in certain peptides, P19, P44–P49, and P60–P69, suggesting the presence of disulfide linkages (Figure 2B and Appendix A). Compared to the peptides in the negative dataset, the thrombin-inhibiting peptides have higher percentages of negatively charged amino acids, E and D and P and G, and lower percentages of positively charged amino acid K, and also the hydrophobic amino acids L, M, V, and A. The highest dipeptide compositions with an average of more than 1.0% in the entire positive dataset are shown in Figure 3A. The composition of dipeptide EE is 2.75%, and of DF, FE, PE, and GD are ~1.7% in the positive dataset, while the corresponding values are 0.13–0.40% in the negative dataset, indicating the importance of these dipeptides on thrombin inhibition. The other dipeptides at significantly higher percentages in the positive dataset are IP, EY, EI, PR, SD, DE, AE, and YL.

Next, we analyzed the distributions of isoelectric point, aromaticity, charge, hydropathy index, and secondary structure proclivities between the positive and negative peptides (Figure 3B). The median isoelectric points of positive and negative peptides were 4.24 and 7.92, respectively, indicating that the positive peptides will be negatively charged at a neutral pH. This is also indicated in the charge distributions in the positive and negative peptides. The median aromaticity values of the positive and negative peptides do not differ significantly. The hydropathy index showed that the positive peptides had a higher distribution of hydrophilic and neutral amino acids compared to the negative peptides, while the negative peptides had a higher distribution of hydrophobic amino acids. The thrombin inhibitory peptides are likely to form more coils compared to the negative peptides, which is consistent with the higher fraction of P and G in the positive dataset.

To investigate the commonality between peptides derived from various sources, we first performed a multiple sequence alignment among all positive peptides, but that approach did not reveal any obvious homology between the sequences. Therefore, we used agglomerative clustering to allow for the evolution of peptide clusters based on underlying similarities between the peptides. Agglomerative clustering is a versatile hierarchical clustering algorithm that does not require specifying the number of clusters in advance. The clustering is performed not only using the amino acid sequence information, but also all the 572 features described above. The peptides are represented using a dendrogram, a tree-like structure that is constructed by merging clusters from the bottom up. The height of fusion on the vertical axis indicates the dissimilarity between the two peptides. Based on the silhouette scores of agglomerative clustering, the optimal number of clusters for the positive set was found to be 48 (Appendix A). While 30 clusters contained only one peptide, 9 clusters contained two peptides, and the largest cluster contained eight peptides (Appendix A).

We constructed a dendrogram to visualize the relationships between peptides in different clusters at various levels (Figure 4). At the lowest level, most of the peptides derived from organisms beloning to the same family cluster together, as indicated by a coloring scheme. These families include: (a) genus *Haemaphysalis* (hard-bodied and bush ticks); (b) genus *Ornithodoros* (soft tick); (c) genus *Anopheles* (mosquito); (d) genus Bothropos (pit viper snake); (e) genera *Hirudo* and *Hirudinaria* (leech); (f) genus *Dipetalogaster* (kissing bug); (g) genus *Crassostrea* (oysters); and (h) genus *Amblyomma* (ixodid tick). Not surprisingly, synthetic peptides clustered together because they were designed from the same template. Further, the peptides belonging to the same cluster inhibited thrombin through similar mechanisms. For instance, clusters (a) and (d) contained peptides that were reported to bind the active site and the exosite II, and to exosite I and exosite II of thrombin, respectively. Most other peptides inhibit thrombin by binding to both active site and exosite I. In contrast, at the highest level, the dendrogram reveals two large clusters: the larger peptides from various natural sources clustered together as one group, while shorter and synthetic peptides clustered together as another group, indicated as top and bottom in Figure 4.

### 3.2. Development of Machine Learning Models for Thrombin Inhibition

The clustering analysis presented above reveals large diversity in the patterns of amino acid sequences in thrombin-inhibiting peptides sourced from different organisms. This suggests that a more sophisticated approach is needed to explore the relationship between these patterns and the inhibitory potential of the peptides, which may guide toward an intelligent drug design. Toward this end, we constructed a machine learning pipeline (Figure 1) consisting of six steps: curation of positive and negative datasets, development of classification model, prediction of new antithrombotic peptides, identification of unique peptides by clustering, development of regression model, and prediction of inhibition constants.

We trained several classification models using the support vector classifier with linear kernel (SVC-L) and with radial bias function kernel (SVC-R), random forest (RF), k-nearest neighbor (KNN), logistic regression (LR), and the XGBoost (XGB) algorithms. We employed 5-fold cross-validation to test the performance of these models, and estimated accuracy, *F*_1_ scores, and MCC. We tuned the hyperparameters of each model to obtain the best MCC scores (Appendix A). We focus on MCC scores for two reasons: First, MCC is widely adopted in peptide classification due to its ability to handle imbalanced datasets and provide a balanced evaluation metric. Second, MCC considers both true positives and true negatives, making it a more comprehensive and reliable performance measure compared to the *F*_1_ score. To evaluate the variance, the model performance was computed for ten different combinations of training, validation, and test sets. As shown in Figure 5A, the MCC scores of all the models for the training set were higher (96.6% to 100%) than those of the validation set (73.5% to 79.7%) and the test set (71.1% to 77.6%). Among the baseline models, SVC RBF achieved relatively good performance with a validation MCC of 79.7% and a test score of 77.5%. XGBoost also showed promising results with a validation MCC of 78.5% and a test score of 77.6%. It is important to note that these scores are the mean values obtained from 10 different sets. It is also worth noting that the standard deviation of the test scores is relatively high compared to the validation scores, indicating some variability in the model’s performance on unseen data. This suggests that the models may not generalize as well to new and unseen samples.

To improve the validation and test MCC scores, we sought to implement feature reduction techniques. A pairwise correlation analysis revealed the existence of a strong correlation (<|0.8|) between 77 features and 54 pairs (Appendix A). Therefore, we performed feature engineering by applying the Recursive Feature Elimination (RFE) algorithm with tuned SVC-L, LR, RF, and XGB models. Sequential Forward Selection (SFS) was performed for the SVC-R and KNN models. The performance of the feature-engineered models is summarized in Figure 5B, and the optimal hyperparameters for these models are listed in Appendix A. After applying RFE/SFS, significant improvements were observed in the validation and test MCC scores for the SVC-L, SVC-R, and LR models. The SVC-L model showed the best performance with a validation MCC score of 83.6% and a test MCC score of 81.1%, followed by the LR model, which showed a validation MCC score of 83.4% and a test MCC score of 82.1%. On the other hand, the KNN model showed a decrease in performance with a validation MCC score of 69.8% and a test MCC score of 64.2%, accompanied by higher standard deviations for both sets. The optimal number of features that gave the best performance for each model is given in Table 1. Based on these results, the SVC-L model with RFE-reduced features (120 features) was selected as the best-performing model.

### 3.3. Prediction of Antithrombotic Efficacy of Peptide Hits

Next, we developed a feature-based regression model to predict the inhibition constants (*K_I_*) of thrombin inhibitory peptides. The original training dataset consisted of 53 peptides with *K_I_* ranging from femtomolar to millimolar. This dataset was filtered to remove ineffective outliers with values larger than 40 μM, which reduced the dataset to 49 peptides. The inhibition constants still spanned eight orders of magnitude, and therefore, we logarithmically scaled the values expressed in nanomolar to the range −5 to +5. We first tested three models, namely, SVM with linear kernel (SVR-L), SVM with RBF kernel (SVR-R), and Lasso regression (Lasso).

Among the baseline regression models, the SVR with RBF kernel demonstrated the best performance with a 5-fold validation RMSE of 1.847 and a test RMSE of 1.541. We applied Sequential Forward Selection (SFS) on all models to improve the model performance.

The SFS algorithms improved the performance of all the models, as seen by a decrease in RMSE (Table 2). Based on the performance of the model on the validation dataset, we chose the SVR with RBF kernel with 125 features as the final regression model. As shown in Figure 6, the predicted *K_I_* values correlate well with ground truth *K_I_* (*R*^2^ = 0.93) over the entire range of concentrations, indicating the robust performance of the model. The predicted values of *K_I_* for the positive dataset, including peptides without known *K_I_* values, are presented in Appendix A.

### 3.4. Prediction of Antithrombotic Activity in Test Peptides

We used the feature-reduced SVM model with the linear kernel to classify 10,743,304 new peptides as those with and without thrombin inhibitory activity. These test peptides represented various sources including fungi (36%), humans (23.3%), eukaryotes (10.8%), bacteria (8.6%), snakes (0.98%), viruses (6.4%), leeches (0.16%), mice (0.36%), and others (13.4%), as shown in the Figure 7A. We extracted the features of these test peptides, as mentioned in the Methods section, and applied the classification model. The model classified 50,325 peptides out of the total test set as positive peptides, or a first pass hit rate of 0.46%. To eliminate false positives, we ranked these positively classified peptides based on their distance from the SVM hyperplane, determined by the decision function of the SVM classifier. A similar ranking of the training set showed that a cutoff value of 0.5 is a good separator to distinguish any erroneously classified false positives (Appendix A). Therefore, using a threshold of 0.5, we found that 15,645 peptides may be considered as true positives with a second pass hit rate of 0.089%. To determine the antithrombotic efficacy of selected peptides, we used the final regression model to predict the *K_I_* values of the 15,645 peptides (Figure 7B). As mentioned earlier, low *K_I_* values indicate a higher affinity for the target and are preferred in terms of thrombin-inhibiting activity. Therefore, to further refine our peptide selection to those that may be truly efficacious, we specifically considered those with log_10_(*K_I_*) <0 (i.e., 1 nM or less) and a decision function ≤ 2, which resulted in a total of 308 peptides, comprising both short and long sequences (Figure 7C).

### 3.5. Clustering of Hits to Identify Unique Peptides

To reduce the set of peptide hits to only unique sequences, we pared down those with overlapping sequences using an agglomerative clustering algorithm with 120 features that were employed in the final classification model. Based on the optimal silhouette score (Appendix A), we found that the 308 peptides were grouped into 126 clusters. The number of clusters showed an exponential distribution with a long tail. While the 56 clusters contained only 1 peptide, 3 clusters contained as many as 11 peptides (Figure 7D). Since each cluster is uniquely represented by a medoid, we obtained 126 medoids. Further, since we are interested only in short peptides for potential translational benefits, we selected medoids with a sequence length of less than 15. After eliminating peptides with >80% sequence similarity, we obtained 59 peptides as the most promising hits from the machine learning-based screening algorithm (Appendix A).

### 3.6. Ranking of Top Hits Based on Binding Scores

To determine the quality of the hits obtained using the machine learning model, we used HPEPDOCK and CABS-dock to dock the 59 peptide sequences with thrombin. HPEPDOCK implements a hierarchical docking protocol with fast conformational sampling of peptide conformations followed by ensemble docking, while CABS-dock uses a replica exchange Monte Carlo algorithm. The docking scores for top-ranked poses from HPEPDOCK are a measure of their binding affinity. Since CABS-dock does not provide such a score, we used the web server PRODIGY to obtain the binding energy for the top-ranked poses generated by CABS-dock. As the most likely candidates for effective thrombin inhibition, we chose 21 peptides that were in the top 50 percentile of the binding/docking scores by both methods (Figure 8). This corresponds to an HPEPDOCK docking score between −224.97 kcal/mol and −166.31 kcal/mol, and CABS-dock derived the binding energy of −11 kcal/mol and −7.5 kcal/mol. The binding affinity computed from the CABS-dock model for these peptides was predicted to be between 9 nM and 1500 nM.

The 21 peptides were 11 to 14 amino acids long (Table 3). The most and least commonly occurring amino acids in the 21 peptides were E, G, D, P, S, H, M, and W, which were also similarly represented in the positive set (Appendix A). Except for ‘FE’, the most occurring dipeptides in the hits were different from the positive set. The charge and polarity distributions were similar to the positive set. Although the average pI of the hits (4.20) was comparable to that of the positive set (4.24), a few peptides (T46, T49, and T55) have a pI of more than 8.5. Next, we ran a BLASTp search on these hits and, based on the percentage sequence identity, we selected the top-ranked matching protein and the source organism. While 50% of the hits matched protein sequences from bacteria, the rest were derived from plants, humans, primates, viruses, fishes, birds, and reptiles. Further, the hit peptides were fragments of enzymes, structural proteins, ribosomal proteins, and synthetic constructs, indicating the diversity of this set.

To obtain the binding sites of the peptides on thrombin, we analyzed the top 10 poses of the 21 thrombin–peptide complexes obtained from CABS-dock using PyMol. We first benchmarked our approach by finding the binding sites on thrombin for positive peptides, hirugen, and avathrin. Consistent with the published reports, our model predicted that these two peptides bind to exosite I [39] and to active site + exosite I [40], respectively (Appendix A).

The docking analysis revealed that the seven peptides bind only to exosite I or to exosite II, two peptides bind only to the active site, and six peptides bind to both the active site and exosite I or exosite II, thus indicating different mechanisms of inhibition of the hits (Table 3). Figure 9 shows the complexes of representative hits with thrombin along with their binding sites.

Next, inspired by the design of bivalent inhibitors such as bivalirudin, we sought to use the peptide hits to design new bivalent inhibitors. To exemplify this approach, we combined peptides that interact with different binding sites on thrombin with varying levels of strength; peptide hits T40 and T33 (both active site binders) were combined with T39, T55 (both exosite I binders), T29, or T27 (both exosite II binders), either at N- or at C-terminal. The bivalent peptides were docked with thrombin and the binding energy was computed. As shown in Appendix A, combining T33 with T39, T55, T29, or T27 resulted in an incremental change in binding energy compared to a single peptide binding to thrombin, and so did combining T40 with T39, T55, or T29. In contrast, as shown in Figure 10A, combining T40 with T27 resulted in a significant change in binding energy for the bivalent peptide compared to either T40 or T27 alone, and the change depends on the N-C concatenation. The T40–T27 concatenation resulted in a significant increase in binding energy, while the T27–T40 concatenation resulted in a significant decrease in binding energy. The docking analysis showed that the favorable binding of the T27–T40 bivalent peptide could be due to a higher binding interaction of this bivalent peptide around the exosite II of thrombin. This ‘wrap-around’ site is possible because of the flexible structure of the T27–T40 peptide, comprising two short helices connected by a flexible loop (Figure 10B). On the other hand, the T40–T27 is a single long helix, the rigidity of which reduces the ability of the bivalent peptide to bind thrombin. For the bivalent peptides T27–T40 and T40–T27, the *K_D_* predicted by PRODIGY are 0.034 nM and 11 nM, and the *K_I_* predicted by our regression model are 0.34 nM and 917 nM, respectively. These values not only confirm the differences in the efficacies of the peptides, but also the qualitative agreement between the mechanism-agnostic ML model and the structure-dependent docking models.

## 4. Discussion

In this work, using a machine learning model pipeline, we have identified features that characterize thrombin-inhibiting activity in peptide sequences and established relationships between these features and their potential for thrombin inhibition. We have discovered, from diverse sources, new peptides with varying levels of antithrombotic activity and varying degrees of sequence homology with known peptide sequences.

This is the first time, to our knowledge, a comprehensive evaluation of known thrombin-inhibiting peptides has been performed. We collected all the available naturally occurring thrombin-inhibiting peptides along with their inhibition constants starting with classical hirudin and other peptides published since 1976 [41]. Our sequence alignment analysis shows that the known, naturally occurring thrombin inhibitor sequences are indeed structurally diverse, with inhibition constants ranging over eight orders of magnitude. This analysis also demonstrated that the varying degrees of effectiveness of antithrombotic peptides depends on the strength of interactions with allosteric or active site interactions or both. Although most known antithrombin inhibitors seem to have originated from hematophagous organisms, this class includes an estimated 15,000 species of arthropods and a large number of leeches and hookworms, and their blood-feeding behavior has evolved independently over six times, suggesting that there are likely to be many more undiscovered peptides [42].

ML-based methods provide much deeper insights into structure-activity relationships than sequence alignment methods. The ML algorithms search higher dimensional spaces for non-physical pattern matching while sequence alignment methods are centered on properties of individual amino acids. Features extracted using ML models corroborated with previous results based on sequence alignment analysis about the characteristics of peptides that are considered important for thrombin binding, namely the presence of negatively charged amino acids (D, E), lower isoelectric point, and hydrophobicity. Interestingly, the most common dipeptide in the positive set ‘EE’ occurred only once in 3 out of the 21 hits, and D/E-containing dipeptides were present in only 8 out of 21 hits, suggesting that thrombin inhibitory activity may also be determined by features defined by the primary sequence. Performing a regression analysis provided additional insights into amino acids and sub-sequences that determine the affinity of the peptide for thrombin. To obtain further insights, we compared the distribution of 120 features in the 21 hits and the 88 positive peptides that were selected by the final classification model (Table 1, SVC-Linear with RFE). In Appendix A, the features are listed along with their relative importance in the model (i.e., weights), and the *p*-value of the distribution of the features between the hit and the positive sets. We found that 87 out of 120 features of the hit peptides were distributed similarly to those of the positive peptides (*p* > 0.05). This observation suggests that while the similarity between the two sets in some of the features is obvious (such as isoelectric point, % of M, and FE), some others are subtler (such as the charge transitions ‘ChargeT13′ and ‘ChargeT23′). Still, others may be unique to hits as these features or dissimilar to those of the positive set (such as EY and PE). This information may be used to design novel peptides that are distinct from the known positive peptides.

This is also the first time, to our knowledge, machine learning models have been applied to the discovery and evaluation of novel thrombin-inhibiting peptide sequences. Previous approaches have focused on the evaluation of small molecule inhibitors based on molecular structures derived from crystallographic information and computationally heavy protein-structure predictions [43,44,45,46,47]. Although structure-based models tend to be more accurate, they also require enormously more, often mechanistic, information compared to ML-based models, which follow an agnostic approach for rapid, high-throughput screening of enormous datasets [48]. The efficiency of machine learning models made it easy to search a vast biological space, resulting in a low hit rate of ~1500 hits per million peptides. Upon screening more than 10 million peptides from various peptide databases for thrombin inhibition activity, our model identified sequences that included both peptides with previously known bioactivity unrelated to thrombin and peptides that do not have any known bioactivity. We identified thrombin-inhibiting activity in antimicrobial, antiviral, antifungal, and anti-inflammatory peptides. Therefore, these peptides may be used for drug repurposing for their anticoagulant properties or may serve to probe the cross-acting role of thrombin in infection or inflammation.

As with most first-of-its-kind studies, this work is not without limitations: (1) We were able to find only 88 unique thrombin-inhibiting peptides, and only 53 with inhibition constants. Further, these inhibition constants were curated from the data generated in different labs with unavoidable variations in the assay conditions. Given that structurally complex peptides of up to 100 amino acids can be reliably and rapidly synthesized on a large scale, we propose that high-throughput experimentation will generate larger databases and improve the model predictions. (2) The hits were tested for sensitivity to thrombin but not for specificity. Therefore, some of the hits may be active against other serine proteases involved in clotting dynamics, including FXa or plasmin, and further analyses for specificity are essential. (3) Both the ML model and the docking model do not classify the type of inhibition (such as reversible/irreversible), although this may be included in the model if additional information is available. (4) The mechanisms of inhibition and interaction sites on thrombin for the peptide were based on top poses that were computed by the model in the absence of any ions or pH changes, which may be critical in a physiological milieu. (5) Experimental validation of these putative hits using in vitro enzyme inhibition assays and in vivo animal models should be performed to confirm the predictive power of the model [49].

Direct Thrombin Inhibitors (DTIs) have a pharmacological advantage over indirect thrombin inhibitors because of their ability to bind both circulating and fibrin-bound thrombin, better efficacy, predictable pharmacokinetics, and fewer off-target effects. Despite these advantages, their usage has been limited to certain indications because of issues with the lack of specific antidotes, bleeding, and clot destabilization. The peptide sequences identified in this work open up the possibility of discovering new DTIs with tunable affinities. Further, the discovery of thrombin inhibitory potential in peptides with known bioactivity, such as antimicrobial, anticancer, and anti-inflammatory, opens up the possibility of drug repurposing for co-morbidity due to thrombotic complications. In vitro inhibition assays of the peptide hits will provide lead candidates with sufficient specificity and sensitivity, which may be advanced to testing in animal models of arterial or venous thrombosis. Lastly, the classification–regression staged model pipeline developed in this work may be readily applied for the discovery of peptides targeting other coagulation proteases such as FXa, FXIa, or plasmin, provided peptide inhibitors and their *K_I_* values are available. Our approach can reduce the turnaround time in drug discovery and provide better quality hits.

## Figures and Tables

**Figure 1 bioengineering-10-01300-f001:**
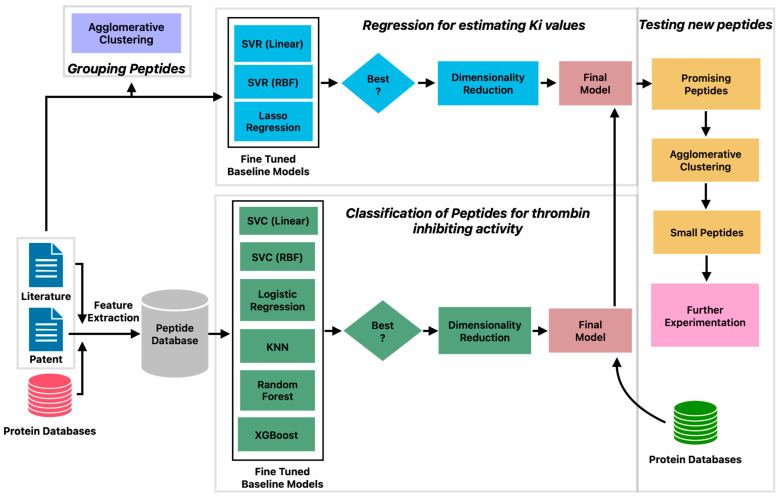
Machine learning pipeline for prediction of peptides with potent thrombin inhibitory activity.

**Figure 2 bioengineering-10-01300-f002:**
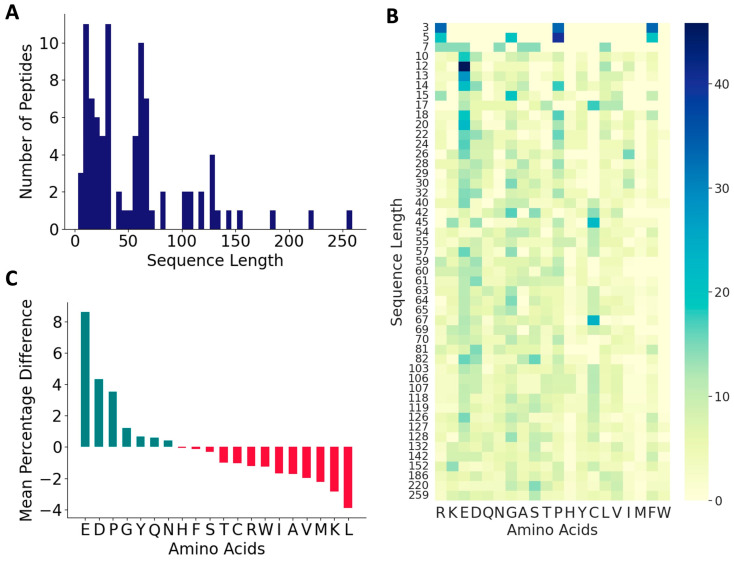
Characteristics of antithrombotic peptides in the positive dataset. (**A**) Distribution of sequence lengths among the 88 peptides in the positive dataset; (**B**) heatmap of various amino acids across peptides of varied sequence lengths in the positive dataset; and (**C**) relative distributions of amino acids.

**Figure 3 bioengineering-10-01300-f003:**
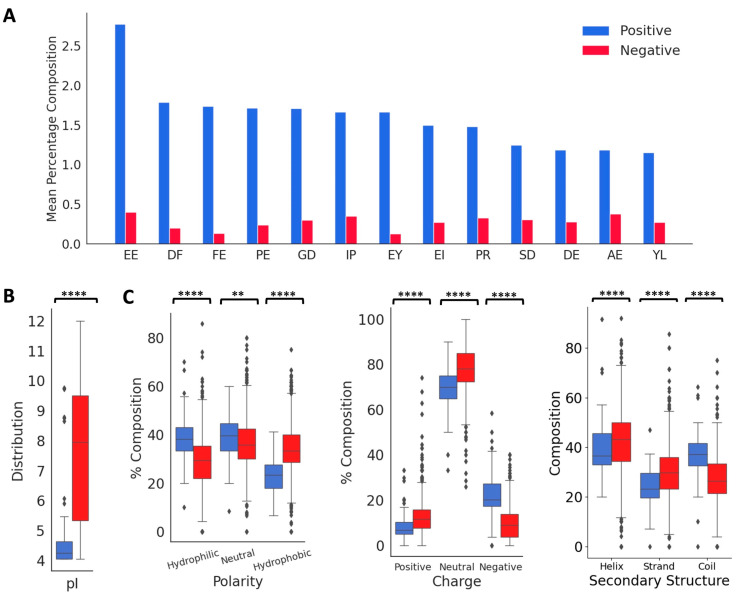
Distribution of key features in the positive and negative datasets. (**A**) Comparison of mean values of dipeptide distributions most represented in the positive dataset. (**B**) Comparison of physico-chemical properties and secondary structure distributions. ‘**’ for *p* < 0.01, and ‘****’ for *p* < 0.0001.

**Figure 4 bioengineering-10-01300-f004:**
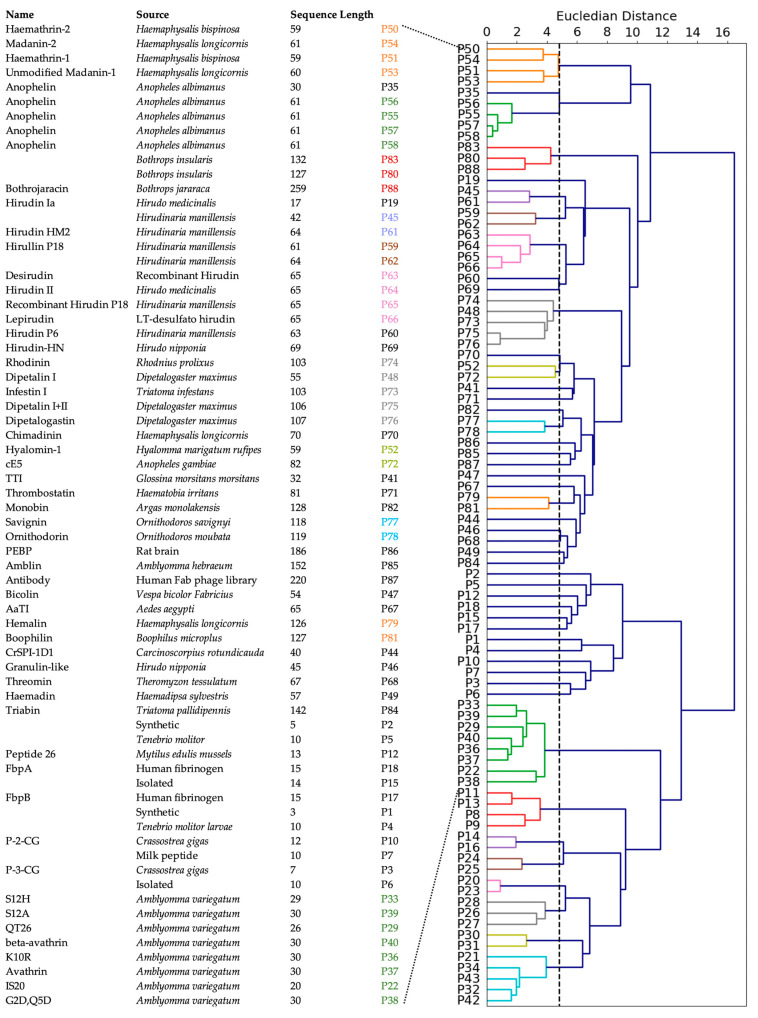
Clustering of positive peptides. Dendrogram visualization of positive peptides reveals that these peptides group in two large clusters. The natural, known peptides are indicated. The peptides at the bottom are synthetic peptides and are not assigned a specific name.

**Figure 5 bioengineering-10-01300-f005:**
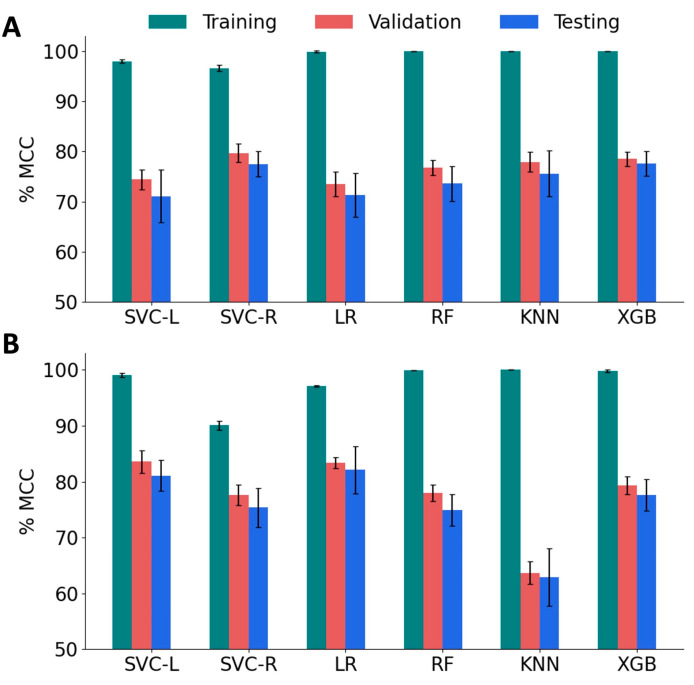
Performance of machine learning models on various sets of features. (**A**) The Matthew’s Correlation Coefficient (MCC) score for baseline classification models, namely, SVM linear kernel (SVC-L), SVM RBF kernel (SVC-R), logistic regression (LR), random forest (RF), and k-nearest neighbors (KNN) and XGBoost (XGB). (**B**) MCC of reduced-feature classification models.

**Figure 6 bioengineering-10-01300-f006:**
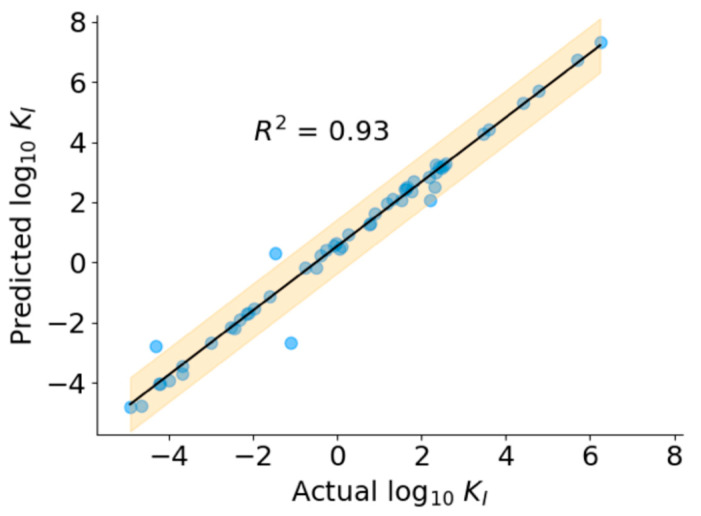
Performance of regression model.

**Figure 7 bioengineering-10-01300-f007:**
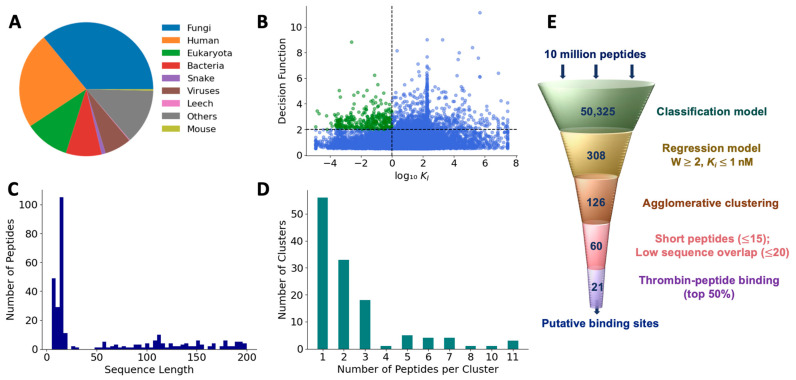
Prediction of antithrombotic peptide hits. (**A**) Distribution of sources of test peptides; (**B**) most efficacious peptides were chosen based on higher decision function and lower *K_I_*; (**C**) sequence length of hits; (**D**) hierarchical clustering of the first-pass hits; and (**E**) filtering pipeline to obtain efficacious peptide hits.

**Figure 8 bioengineering-10-01300-f008:**
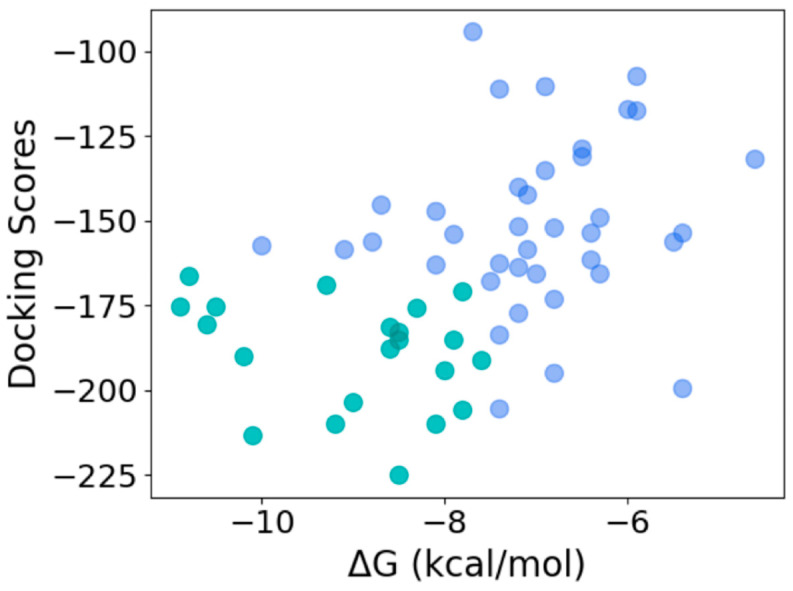
Selection of most efficacious hits at the intersection of the top 50 percentile of docking scores from HPEPDOCK and binding energy obtained from CABS-dock-derived structures (top 50 and bottom 50 percentiles shown in green and blue, respectively).

**Figure 9 bioengineering-10-01300-f009:**
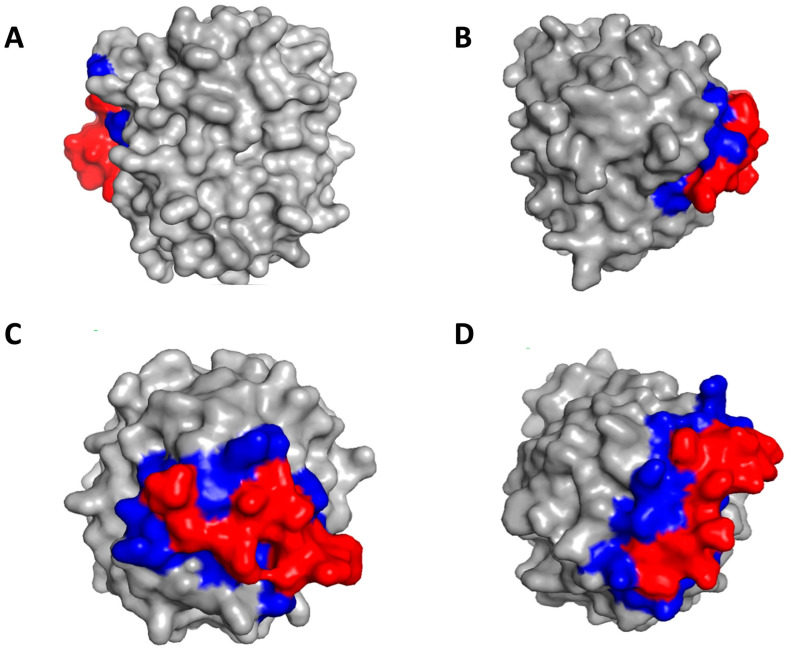
Identification of binding sites on thrombin by peptide hits. (**A**) Peptide T29 binds to exosite II; (**B**) Peptide T43 binds to exosite I; (**C**) Peptide T40 binds to active site; (**D**) Peptide T45 binds to exosite I and active site. The red color denotes the peptide and the blue color denotes the binding residues on thrombin.

**Figure 10 bioengineering-10-01300-f010:**
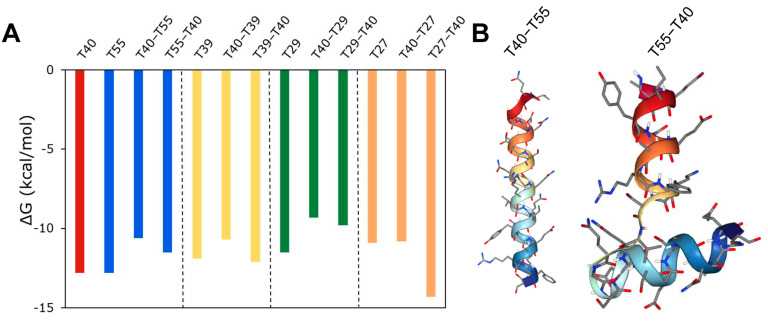
Design of a bivalent peptide. (**A**) Free energy change in binding of thrombin to bivalent peptide composed of T40 peptide combined with either T55, T39, or T27 peptides in either N-C or C-N concatenation. (**B**) 3D conformation of T40–T27 and T27–T40 peptides.

**Table 1 bioengineering-10-01300-t001:** Optimal number of features selected from RFE/SFS.

Model	Feature Reduction Method	Number of Features
SVC Linear	RFE	120
SVC RBF	SFS	314
Logistic Regression	RFE	54
Random Forest	RFE	508
KNN	SFS	257
XGBoost	RFE	32

**Table 2 bioengineering-10-01300-t002:** Comparison of performance of regression models before and after SFS optimization.

Model	Stage	Log Training RMSE	Log Validation RMSE	Log Test RMSE
SVR with Linear Kernel	Baseline	0.715	1.951	1.47
	SFS with 51 features	0.778	1.149	1.221
SVR with RBF Kernel	Baseline	0.279	1.847	1.541
	SFS with 125 features	0.2	1.114	1.06
Lasso Regression	Baseline	1.14	1.887	1.802
	SFS with 28 features	1.388	1.728	1.107

**Table 3 bioengineering-10-01300-t003:** Binding characteristics of peptide hits.

Peptide	Sequence	Source	*K_D_* (nM)	Docking Scores	Binding Residues	Binding Sites
T49	QGNRKTTKEGSNDL	*Homo sapiens* (cytokine-dependent hematopoietic cell linker isoform X1)	9	−175.365	S20, D21, A22, E23, I24, G25, M26, P28, K70, H71, E80, D116, Y117, I118, Y134, K135, R137, V158, N159, E185, K202, S203, P204, N205, R206, W207	Exosite 1
T34	EYEEVEASPEKET	*Meleagris gallopavo* (tubulin beta-1 chain)	12	−166.309	I47, S48, W51, K87, Y89, I90, H91, P92, R93, L105, K107, K109, K110, P111, V112, C122, L123, R126, E127, F232, K236, K240	Exosite 2
T45	SGEGSFQPSQQNPQ	*Triticum aestivum* (gliadin peptide)	16	−180.56	F34, R35, K36, S37, P38, Q38A, E40, R67, K70, H71, R73, T74, R75, Y76, E77, R77a, N78, I79, W141, N143, L144, Q151, P152, S153	Active site and Exosite 1
T56	ARATAETDATANRG	*Mycobacterium tuberculosis* (prophage protein)	20	−175.107	E23, I24, K36, P38, K70, H71, S72, R73, T74, R75, E77, E80, S153, V154	Exosite 1
T52	EPTTEDLYFQSDND	M13 helper phage (pIII)	31	−189.841	N98, D100, R101, T147, R173, R175, T177, E217, R232	Active site and Exosite 2
T33	IYRFEPSKFIGE	*Nymphaea colorata* (unnamed protein)	37	−213.129	E39, L40, R93, E98, N143, S171, R173, I174, R175, I176, E192, E217, A221D	Active site
T39	ACENEDFEGIPGEA	*Homo sapiens* (hirugen, synthetic construct)	150	−168.785	S20, D21, A22, E23, I24, G25, M26, P28, W29, I68, G69, K70, I79, E80, K81, A113, F114, S115, D116, K135, G149a, K149b, V157, V158, N159, E184a, G186c, K186c, K202, S203, P204, R204a	Exosite 1
T57	FEFEFEPGGGRGDS	*Spirochaetales bacterium* (SpoIIE family protein phosphatase)	170	−209.705	K36, S37, P38, Q39, E40, W60a, S72, R73, T74, R75, Y76, R97, E98, N99, N143, L144, K145, W147, T148, Q151, S153, C191, D221a	Active site and Exosite 1
T54	RYEVRAELPGVDPD	*Mycobacterium tuberculosis* (erythromycin esterase)	240	−203.537	E23, M32, R35, K70, H71, R73, T74, R75, Y76, V154, Q156	Exosite 1
T27	VQIYEEARKFS	*Potamochoerus porcus* (DEAD-box protein 3)	480	−187.722	H91, P92, R93, Y94, L99, D100, R101, D125, I176, T177, N179, H230, V231, F232, R233, L234, W237, I238, I242, D243	Exosite 2
T44	GNTRTAESGDEDFF	*Eubacteriales bacterium* (transglycosylase domain-containing protein)	530	−181.246	R35, P38, Q39, E40, R67, K70, H71, S72, R73, T74, Y76, E77, R78, N79, E80, G142, N143, L144, Q151, P152, S153, V154, E192	Active site and Exosite 1
T55	NRLVQNPPKKFSGE	*Burkholderia* sp. Bp9140 (hypothetical protein)	610	−224.973	S20, D21, A22, E23, Q39, V67, H71, S72, T74, Y76, S116, Y117, K135, W141, A149V, S153, V154, L155, V157, N158, E185, R187, K202, S203, P204, R206	Exosite 1
T31	AEYETVQNSFNQ	*Cellvibrio fibrivorans* (cellulase family glycosyl hydrolase)	630	−185.041	P92, R93, W96, N98, L99, D100, R101, D102, I103, R126, A129A, B129S, Q131, E164, R175, I176, T177, D178, N179, H230, F232, R233, K236, Q244	Exosite 2
T46	SSGSVGESSSKGPR	*Pan pansicus* (cytokeratin-10)	630	−182.793	E23, I24, G25, F34, R35, S37, P38, Q39, E40, L41, D60W, K70, H71, S72, R73, N79, E98, N99, D116, N143, L144, K145, P152, S153, L155, Q156, E192, W215, G216, E217	Active site and Exosite 1
T40	VQGSDQSDSANVQR	*Hoeflea* sp. (UDP N-acetylmuramate L-alanine ligase)	770	−175.557	I23, F34, R35, K36, S37, P38, Q39, E40, L42, R73, W140, N143, L144, E146, C149V, P152, S153, E192, E216, G218, C219, D220, R221	Active site
T41	NDDEDPKSHRDPSN	FGF-4 synthetic construct	1200	−209.886	I24, G25, Q30, K70, R78, N79, I80, E80, K81, I82, K107, L108, K109, K110, P111, V112, F114, Y117, I118, H119	Exosite 1
T32	GEKPDEFESGSP	*Poecilia Mexicana* (ribosomal protein S7)	1300	−194.088	R101, R126, T128, A129A, S130, L132, Q133, E164, R165, K169, D178, N179, M180, S203, P204, F205, H230, R233, K236	Exosite 2
T42	RGNNDIGSGFNDDP	*Cellulomonas soli* (glycosyl transferase)	1600	−185.149	N95, W96, E97, N98, L99, D100, V163, P166, K169, D170, S171, T172, I174, R175, I176, Y184a, K184b, E185, E217, R221b, D222, G223, K224	Exosite 2
T48	GIGPKFQHSGGEPP	*Mycobacterium tuberculosis* (prophage protein)	1800	−205.784	I90, H91, R93, Y94, N95, N99, L100, D100, R173, I174, R175, I176, F227, V241, I242, F245, E246	Exosite 2
T29	MEEGPSDPGSRS	*Mogibacterium* sp. (haloacid dehalogenase-like hydrolase)	1800	−170.831	I24, G25, Q30, K70, R78, N79, I80, E80, K81, I82, K107, L108, K109, K110, P111, V112, F114, Y117, I118, H119	Exosite 2
T43	HGEGTFTSDLSKQM	*Heloderma suspectum* (exendin 4 venom)	2500	−190.877	E23, I24, G25, M26, E39, K70, H71, E77, R77a, N78, I79, D116, I118, H119, N143, S153, V154, Q156	Exosite 1

## Data Availability

The data presented in this study are available in the Appendix A.

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
