# Peer review of "Prospection of Peptide Inhibitors of Thrombin from Diverse Origins Using a Machine Learning Pipeline"

_bioengineering, 2023, doi:10.3390/bioengineering10111300_

Round 1

Reviewer 1 Report

Comments and Suggestions for Authors

See attatched file. There are some major and some minor concerns that need to be addressed.

Comments on the Quality of English Language

There are some phrases where a word is missing, e.g. on page 20: "This is also the first time, to our knowledge, THAT machine learning models" Pleas carefully check.

Author Response

Reviewer 1

Major concern: 

1) The origin of the 10,743,304 peptides is not properly described. It should be clarified whether these peptides are "true" peptides (and produced as such, e.g. antimicrobial peptides, atrial natriuretic peptides or hirudins) or are (random) parts of larger proteins.

The peptides were collected from the NCBI and UniProt databases. We searched for the source as one of ‘Fungi’, ‘Bacteria’, ‘Snake’, ‘Leech’, ‘Humans’, ‘Mouse’, ‘Eukaryota’, ‘Viruses’, and were filtered to a sequence length between 5 to 200 amino acids. Peptides in the sequence range of 5 to 15 amino acids were collected independent of the source. The peptides could be true peptides or random fragments of large proteins. This information is included in the text (p. 6 line 2-6). 

2) In that context: the source of the peptides should be carefully evaluated. In Table 3, 14 out of the 21 peptides are of an "unknown source". Using a simple Re-BLAST against the NCBI protein database solves the problem and uncovers the sources of almost all "unknown" peptides. Only three examples: Peptide T49 comprises the amino acid residues 19-32 of the human cytokine-dependent hematopoietic cell linker isoform X1, peptide T46 comprises the amino acid residues 570-583 of the human keratin 10 and peptide T43 is identical to the first 14 amino acid residues of exendin-4, a venom component of Heloderma suspectum, the Gila monster.

We thank the reviewer for this suggestion. We ran a BLASTp search on all 21 hits and based on percentage identity, we selected the top ranked protein and its source. This information is updated in the ‘source’ column in Table 3. While 50% of hits were from different bacteria, the rest included plants, humans, primates, viruses, fishes, birds, and reptiles. Further, the peptides were fragments of enzymes, structural proteins, ribosomal proteins, and synthetic constructs (Table 3, p 21). 

3) The features of all the 21 peptides that are listed in table 3 should be provided and compared to the features that where "considered to be important for thrombin binding, namely the presence of negatively charged amino acids (D, E), lower isoelectric point, and dipeptides containing D and E". Interestingly, most of the 21 peptides indeed comprise a pI value of about 4, but there are certain exceptions. E.g., peptide T49 has a pI value of about 8.6 and T55 of about 10.

We thank the reviewer for this suggestion too. We have provided the amino acid distribution, pI, charge, polarity, and most represented dipeptide composition of the 21 peptide hits in new Fig S7. The most and least commonly occurring amino acids in the 21 peptides were E, G, D, P,  S and H, M, and W; which were also similarly represented in the positive set. However, the most occurring dipeptides in the hits were different from the positive set. The charge and polarity distributions were similar to the positive set; and the average pI of the hits (4.20) was also comparable to that of the positive set (4.24). Interestingly, we found that peptides T46, T49, and T55 have pI more than 8.5. This information is included now (p 19, lines 2-12, and Fig. S7) 

4) Unfortunately, there is no experimental verification whether or not the 21 peptides are actually thrombin inhibitors. A functional characterization of one or two examples would greatly enhance the validity of the whole concept.

We would like to thank the reviewer for the suggestion. Given the significant cost for peptide synthesis (including hits, positive and negative controls), we plan to pursue experimental validation and possible modifications to the hits in a follow up work. 

Minor concerns: 

1) The term "habitat" is misleading and should be replaced by "origin". Habitat is an ecological term and describes the whole biotic and a-biotic factors that are present in a particular area. 

We thank the reviewer for this note, this change is incorporated in the revision (p 1).

2) In Figure 2B and the respective part of the text, the striking over-representation of cysteine residues in the range of 63-67 (probably hirudins), 42-45 and 17 amino acid residues should be mentioned. 

This change is incorporated in the revision (p 10, lines 10-11).

3) In the Introduction, the abbreviations MCC and RMSE are not introduced. 

The reviewer may have meant Abstract. This change is incorporated in the revision (p 3, lines 11 and 13).  

4) Hirudin was identified in Hirudo medicinalis, but not in Hirudin hirudin (there is no leech with that species name).

We apologize for this oversight. This change is incorporated in the revision (p 34, line 32).

5) The reference list is not yet in an uniform style.

The reference list has been updated to a uniform style. 

6) There are some double spaces left in the text.

The formatting error has been corrected.  

7) There were no line numbers.

The line numbers are added to the revision. 

8) I had no access to the supplementary files. 

We apologize for this issue. We had uploaded the SI along with the main manuscript during submission. We have uploaded the modified version of the SI, and would request to let the editor know if there is an issue during the review of the resubmission manuscript.

Reviewer 2 Report

Comments and Suggestions for Authors

This article presents the use-case for a two-staged machine learning pipeline to identify and rank peptide sequences based on their effective thrombin inhibitory potential. This is of relevance to increase the effectiveness of direct thrombin inhibitors (DTI) requisite for the treatment of various cardiovascular diseases/complications.

The manuscript provides valuable insights to the community on the applicability of ML pipeline to predict thrombin inhibitory activity of peptides from their properties. The two-stage pipeline identifies hits with thrombin inhibitor activity and regression models that predict the level of activity of the peptide hits from a large peptide dataset. The models prioritise the QSAR properties which are influential to thrombin inhibition and then ranks the binding affinity to thrombin as determined by the molecular modeling of their functional interactions.

Overall, the article is well written and coherent; the authors can consider the following points to further enhance the quality of the manuscript with clarification on:

Comments:

·       Section 2.2: Unclear use of ‘n-dimensional’ related to the vectors. Is this referring to vectors or matrices? The shape/type of input is not clear to the reader.

·       Section 2.3: The size of the set is not clear? Only the testing set is explicitly written. Also, what is the reasoning behind the 60/20/20? Were different combinations tested?

·       Section 3: Why not use a neural network as an additional method for classification/regression?

·       page-2, Introduction, 4th paragraph, refine sentence to read as follows, with added citation:"Machine learning algorithms leverage large chemical datasets for predictive modeling and pattern recognition, including the prediction of the structure, properties and activities of peptides including intra- and inter-peptide interactions [12b]." [12b] article title An ab initio exploratory study of side chain conformations for selected backbone conformations of N-acetyl-l-glutamine-N-methylamide 2001

Author Response

Reviewer 2

This article presents the use-case for a two-staged machine learning pipeline to identify and rank peptide sequences based on their effective thrombin inhibitory potential. This is of relevance to increase the effectiveness of direct thrombin inhibitors (DTI) requisite for the treatment of various cardiovascular diseases/complications.

The manuscript provides valuable insights to the community on the applicability of ML pipeline to predict thrombin inhibitory activity of peptides from their properties. The two-stage pipeline identifies hits with thrombin inhibitor activity and regression models that predict the level of activity of the peptide hits from a large peptide dataset. The models prioritise the QSAR properties which are influential to thrombin inhibition and then ranks the binding affinity to thrombin as determined by the molecular modeling of their functional interactions. 

Overall, the article is well written and coherent; the authors can consider the following points to further enhance the quality of the manuscript with clarification on: 

Comments:

Section 2.2: Unclear use of ‘n-dimensional’ related to the vectors. Is this referring to vectors or matrices? The shape/type of input is not clear to the reader.

We apologize for this miswording. The vector refers to a single peptide. The term ‘n-dimensional’ is replaced by ‘n-element’. (p 6, lines 13, 20, 31). 

Section 2.3: The size of the set is not clear? Only the testing set is explicitly written. Also, what is the reasoning behind the 60/20/20? Were different combinations tested?

The training set consisted of 528 samples, the validation set of 176 samples, and the testing set of 176 samples, which served as out-of-sample test data. We utilized the conventional 60/20/20 split for training, validation, and testing, a widely-accepted practice in machine learning [10.1093/nar/gkad303; 10.1007/s00779-019-01248-7]. Given our negative-to-positive sample ratio of 9:1, this division ensures an adequate presence of positive samples in all datasets. Other choices such as 70/15/15 or 80/10/10 splits would reduce the number of positive peptides in the validation and testing sets to 13 and 8, respectively. Therefore, the 60/20/20 split not only aids the model in capturing the intricacies of the data but also allows for a rigorous evaluation. This  information is included in the revised manuscript (p 7, lines 4-6). 

Section 3: Why not use a neural network as an additional method for classification/ regression? 

We had performed neural networks as one of the methods of classification. However, the neural network model did not perform well and exhibited overfitting issues. The validation score yielded a MCC of 0.65 which is too low compared to the models shown in Fig. 5. In hindsight, this was expected as neural networks typically perform well with larger datasets; and hence was not included in the manuscript. 

page-2, Introduction, 4th paragraph, refine sentence to read as follows, with added citation:"Machine learning algorithms leverage large chemical datasets for predictive modeling and pattern recognition, including the prediction of the structure, properties and activities of peptides including intra- and inter-peptide interactions [12b]." [12b] article title An ab initio exploratory study of side chain conformations for selected backbone conformations of N-acetyl-l-glutamine-N-methylamide 2001

We would like to thank the reviewer for pointing to this reference. This reference primarily deals with side-chain-backbone interaction particularly in glutamine, and we could not see the implementation of or direct impact in ML models demonstrated in this work. Therefore, to capture reviewer’s suggestion, we modified the statement to include the phrase “... based on side chains”, and have included four references that discuss and demonstrate the use of side chain properties and interactions on protein design and prediction (p 5, line 1, references 13-16).

Reviewer 3 Report

Comments and Suggestions for Authors

This paper entitled: Prospection of Peptide Inhibitors of Thrombin From Diverse Habitats Using a Machine Learning Pipeline is an excellent paper and contains new intersting method.

The paper has to be published after minor revision.

For example, the different equations in the manuscript have to be numbered. and some physicochemical date should be added

Author Response

This paper entitled: Prospection of Peptide Inhibitors of Thrombin From Diverse Habitats Using a Machine Learning Pipeline is an excellent paper and contains new interesting method.

The paper has to be published after minor revision.

For example, the different equations in the manuscript have to be numbered. and some physicochemical date should be added

The equation numbers have been added. A CSV file containing all the features, including the physicochemical properties, of the positive dataset has been uploaded as a supplement.

Round 2

Reviewer 1 Report

Comments and Suggestions for Authors

Manuskript-ID:   bioengineering-2637560_V2

Journal:               Bioengineering

Title:                     Prospection of Peptide Inhibitors of Thrombin From Diverse Origins Using a Machine Learning Pipeline

Authors:              Nivedha Balakrishnan, Rahul Katkar, Peter V. Pham, Taylor Downey, Prarthna Kashyap, David C. Anastasiu and Anand K. Ramasubramanian

I was a little surprised that there was no point-by-point reply to the questions and concerns. However, the authors addressed all of the minor concerns and the first two major concerns, and I appreciate that. Unfortunately, the major concerns 3 (match between feature predictions and results, e.g. only 9 out of the 21 peptides indeed contain D/E dipeptides) and especially 4 (experimental verification of thrombin inhibition) received less attention by the authors. I find it a pity that the last point was completely ignored, both in the original, and the revised version of the manuscript. The experimental proof of a concept It is the final and crucial step. Without the proof even the title of the manuscript is misleading: there is no "Prospection of Peptide Inhibitors of Thrombin...", there is only a "Prospection of putative/likely/maybe Peptide Inhibitors of Thrombin...".

One short note: lane 116: "..., and the results were filtered..."

Author Response

I was a little surprised that there was no point-by-point reply to the questions and concerns. However, the authors addressed all of the minor concerns and the first two major concerns, and I appreciate that.

We had provided a point-by-point response in the space provided in the portal along with changes that were made in the manuscript with appropriate line and page numbers of the PDF. We are truly sorry that our response was not recorded properly. We are addressing the two points raised by the reviewer in their second review.  We are also uploading the PDF of the response. 

Concern 1.

Unfortunately, the major concerns 3 (match between feature predictions and results, e.g. only 9 out of the 21 peptides indeed contain D/E dipeptides).

 To obtain insights into the features in the predicted peptides and their similiarity to positive peptides, we compared, between the 21 hits and the 88 positive peptides, the distribution of 120 features that were selected by the final classification model (Table 1, SVC-Linear with RFE).  In Table S7, the features are listed along with their relative importance in the model (i.e., weights) and the p-values of the distribution of the features between the hit and the positive sets. We found that 87 out of 120 features of the hit peptides were distributed similar to those of the positive peptides (p>0.05), and the rest 33 features were different between the two sets. A closer look at this demarcation suggests that while similarity between the two sets in some of the features are obvious (such as isoelectric point, % of M, and FE), some others are subtler (such as the charge transitions ‘ChargeT13’ and ‘ChargeT23’). Still others may be unique to hits as these features or dissimilar to those of the positive set (such as EY and PE). Therefore, we think that although D/E peptides may not be present in the hits, other features deemed important by the ML model may contribute to their antithrombotic property. This is an important advantage of ML model is that it may unravel features that may be hidden and not obvious, and may help in designing novel peptides. We have included this information in the Discussion section, along with Table S2 listing the CTD features. We would like to thank the reviewer for bringing up this comparison which has provided additional insights. (p. lines 15-18; lines 20-30)

Concern 2.

I find it a pity that the last point was completely ignored, both in the original, and the revised version of the manuscript. The experimental proof of a concept It is the final and crucial step. Without the proof even the title of the manuscript is misleading: there is no "Prospection of Peptide Inhibitors of Thrombin...", there is only a "Prospection of putative/likely/maybe Peptide Inhibitors of Thrombin...".

   We would like to thank the reviewer for the suggestion. Unfortunately, since we do not have funding to cover the significant cost for peptide synthesis (including multiple hits, positive and negative controls) and assay costs, we plan to pursue experimental validation and possible modifications to the hits in a follow up work. Correspondingly, we have added the adjective ‘putative’ in the abstract, and conclusion, and highlighted the need for in vitro validation and in vivo experimentation to move them forward as true DTI candidates. (p. 3, line 18; p. 25, lines 19-20).

One short note: lane 116: "..., and the results were filtered..."

            This change has been incorporated.
